# Learning Information Propagation in the Dynamical Systems via Information Bottleneck Hierarchy

## Abstract

Extracting relevant information, causally inferring and predicting the future states with high accuracy is a crucial task for modeling complex systems. The endeavor to address these tasks is made even more challenging when we have to deal with high-dimensional heterogeneous data streams. Such data streams often have higher-order inter-dependencies across spatial and temporal dimensions. We propose to perform a soft-clustering of the data and learn its dynamics to produce a compact dynamical model while still ensuring the original objectives of causal inference and accurate predictions. To efficiently and rigorously process the dynamics of soft-clustering, we advocate for an information theory inspired approach that incorporates stochastic calculus and seeks to determine a trade-off between the predictive accuracy and compactness of the mathematical representation. We cast the model construction as a maximization of the compression of the state variables such that the predictive ability and causal interdependence (relatedness) constraints between the original data streams and the compact model are closely bounded. We provide theoretical guarantees concerning the convergence of the proposed learning algorithm. To further test the proposed framework, we consider a high-dimensional Gaussian case study and describe an iterative scheme for updating the new model parameters. Using numerical experiments, we demonstrate the benefits on compression and prediction accuracy for a class of dynamical systems. Finally, we apply the proposed algorithm to the real-world dataset of multimodal sentiment intensity and show improvements in prediction with reduced dimensions.

## 1 Introduction

The use of machine learning for making inference and prediction from the real-world data has shown unprecedented growth. There exist a plethora of approaches for complex system (CS) modeling (e.g., multi-input multi-output state space identification (Stoica & Jansson, 2000), expectation maximization (EM) (Martens, 2010), regularization (Chiuso, 2016), graphical models (Meinshausen & Buhlmann, 2006), combined regularization and Bayesian learning (Fox et al., 2008; 2011; Bonettini et al., 2015), kernel-based regularization (Pillonetto & Chiuso, 2015)). With the increase in the size of data, the complexity of the accurate models also increases, making inference and predictions slower. The major challenges of the upcoming era, hence, are likely to deal with the massive and diverse data sources, and still making quick decisions. Therefore, the compact modeling of time-varying complex systems[1] is a challenging task and appealing for more investigation. The real-world data has complex inter-dependencies across spatial and temporal dimensions. We aim to identify such dependencies and carefully construct a *compact* representation of the given CS model while still ensuring accurate predictions. We do so by performing a soft-clustering of such inter-dependencies to preserve only the relevant information. For the CS model in the form of a dynamical system, we additionally argue that similar to the data, the most relevant information also gets transformed at each hop in an *alternate dynamical system*. From a bird's-eye view, we track

---

[1] In this work, we interpret complex systems (networks) (Barrat et al., 2008; Gao et al., 2014) as graphs comprising of nodes interacting spatially and temporally, i.e. both inter and intra dependence, with node activities available in the form of time-series.

how the most relevant information propagates across the given dynamical system. We represent this propagation via an alternate dynamical system (compact model) and develop an unsupervised learning technique of such process.

The most relevant work in this regard is information bottleneck (IB) principle (Tishby et al., 2000). For fixed two random variables, it performs a soft-clustering to compress one variable while predicting another, given the joint probability distribution. The IB has been successfully applied to speech recognition (Hecht & Tishby, 2005), document classification (Slonim & Tishby, 2000), gene expression (Friedman et al., 2001) and deep learning (Tishby & Zaslavsky, 2015), etc., and it has shown good performance. In contrast, we aim to learn a *dynamics of the soft-clustering* across the given dynamical system, and propose a general optimization framework to study the trade-offs between compactness and the resulting accuracies.

The problem statement addressed in this work is: *Given* a dynamical system, we aim to develop a compact model by *learning* the dynamics of the soft-clustering in an unsupervised manner, or alternate dynamical process, through Information Bottleneck hierarchy (IBH). The main contributions of the present work are as follows: (i) By learning the dynamics of the soft-clustering, we propose an alternate compact dynamical system of the given process, with emphasis on the prediction accuracies. (ii) We formulate a novel optimization setup, *compact perception problem*, and characterize general solution to the information theoretic problem. (iii) We quantify how most relevant information about future gets transformed at each hop in the alternatively designed dynamical system.

A brief mention of the mathematical notations is provided in the next part.

## 1.1 DEFINITIONS AND NOTATIONS

In this manuscript, we use capital letters to denote random variables (RVs), and lowercase letters are used for the realizations. The bold letters are used for multi-variate RVs. For a RV $X$, with little abuse of notation, we denote the probability mass function $p_X(x)$ as $p(X)$, unless specified otherwise. The expectation operator is denoted as $\mathbb{E}[.]$. A Gaussian distributed multi-variate RV is denoted as $\mathbf{X} \sim \mathcal{N}(\boldsymbol{\mu}_\mathbf{X}, \boldsymbol{\Sigma}_\mathbf{X})$, where $\boldsymbol{\mu}_\mathbf{X}$ and $\boldsymbol{\Sigma}_\mathbf{X}$ are the mean vector and covariance matrix, respectively. Next, we present a few information theoretic definitions relevant to this work.

**Definition 1.** *The Kullback-Leibler (KL) divergence (Cover & Thomas, 1991) between two probability mass functions $p(\cdot)$ and $q(\cdot)$ is written as*

$$D_{KL}(p||q) = \sum_x p(x) \log \frac{p(x)}{q(x)}. \tag{1}$$

The KL divergence is in general not symmetric, and $D_{KL} \geq 0$ with $D_{KL}(p||q) = 0$ if and only if $p = q$. Using (1), the mutual information between two RVs $X$ and $Y$ is defined as $I(X;Y) = D_{KL}(p(X,Y)||p(X)p(Y))$. Next, we state the problem statement addressed in this work.

## 2 PROBLEM FORMULATION

Complex systems consist of a large number of interacting dynamic components. In practical situations, we have high-dimensional time-series (node activities) having complex spatial and temporal correlations. The challenge lies in identifying such complicated inter-dependencies and recover the true dimensionality of the system in an unsupervised manner. We equivalently call it as identifying the compact model while preserving the relevant information across spatio-temporal states. Under stationary assumptions, (Tishby et al., 2000) proposed the IB to process information compactly. For the sake of completeness, we provide a brief overview of the IB principle.

## 2.1 INFORMATION BOTTLENECK APPROACH

The IB compresses a variable $X$ into a new stochastic variable $B$ via soft-clustering, while maintaining as much information as possible about another variable of interest $Y$. The variable $B$ operates to minimize the information compression task and to maximize the relevant information. Provided that the relevance of one RV to another is measurable by an information-theoretic metric (i.e., mutual information), the trade-off can be written as the following variational problem

$$\min_{p(B|X)} I(X;B) - \beta I(B;Y), \tag{2}$$

where $\beta$ controls the trade-off between the tasks mentioned above. Hence, the variable $B$, solving the minimization problem (2), encodes the most informative part from the input $X$ about output $Y$.

Inspired by the IB concept and its internal predictive coding, we propose a causal inference framework for discrete-time stochastic dynamical systems. More precisely, we label two consecutive dynamic states $\mathbf{X}_k$ and $\mathbf{X}_{k+1}$ to be the input and the output of the bottleneck $\mathbf{B}_k$, respectively. Hence, $\mathbf{B}_k$ carries the most informative part (relevant information) from the past about the future. Next, we formalize this idea to create a sequence of bottlenecks (or dynamics of soft-clustering) to compactly and accurately represent the given dynamical system.

## 2.2 LEARNING INFORMATION PROPAGATION: SETUP

We consider a stochastic dynamical system, involving large number $n$ of random processes $\mathbf{X} = [X_1, X_2, ....X_n]$, interacting and evolving in time. A fundamental problem for decision making or prediction is to learn a compressed representation of the system dynamics from high-dimensional data (i.e., system identification). The higher-order dependencies hiding behind the high-dimensions make this task challenging, and it needs sophisticated techniques to extract meaningful information for an appropriate application. In this paper, we propose a framework that focuses only on the dynamics of the relevant information, by learning an alternate representation of the given process.

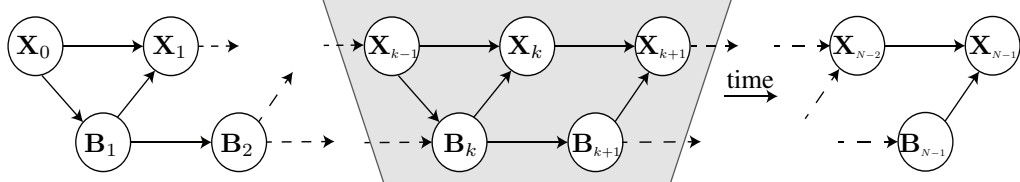

Figure 1: A $N$-length stochastic dynamical system with corresponding IBH in parallel. We study the shaded 3-hop process in isolation, i.e., given three consecutive states $\mathbf{X}_{k-1}$, $\mathbf{X}_k$ and $\mathbf{X}_{k+1}$ of the input process, the stochastic variables $\mathbf{B}_k, \mathbf{B}_{k+1}$ represent dynamics of the alternatively designed process to capture the relevant information.

Adopting an information theoretic representation, we aim to determine $\mathbf{B}_k$ and $\mathbf{B}_{k+1}$ jointly since they provide *compressed predictive information* about the system dynamics. Figure 1 summarizes our objectives, and we study the shaded region (3 states) of the dynamical system in isolation, i.e. without the influence of any other RVs in the complete system. An argument to generalize this study to any $N$-length is provided later in Section 3.1. We determine the stochastic variable $\mathbf{B}_k$ that not only compresses $\mathbf{X}_{k-1}$ as much as possible while preserving the relevant information about $\mathbf{X}_k$, but also delivers this information to $\mathbf{B}_{k+1}$. The variable $\mathbf{B}_{k+1}$ quantifies the meaningful information about the state of the system at time $k + 1$, building upon the compressed information received via $\mathbf{B}_k$, thus forming a dynamics of the relevant information.

By construction, the new mapping $\mathbf{B}_k$ is designed from $\mathbf{X}_{k-1}$ to preserve the consistent information about $\mathbf{X}_k$, therefore, given $\mathbf{X}_{k-1}$, $\mathbf{B}_k$ and $\mathbf{X}_k$ are independent. Similarly, the mapping $\mathbf{B}_{k+1}$ is independent of $\mathbf{X}_{k+1}$ given $\mathbf{B}_k$. Since, $\mathbf{B}_{k+1}$ carries information from $\mathbf{B}_k$ which is compressed representation of $\mathbf{X}_{k-1}$, then given $\mathbf{B}_k$, $\mathbf{X}_{k-1}$ and $\mathbf{B}_{k+1}$ are also independent. With this framework, the following Markov chains are considered: $\mathbf{X}_k$–$\mathbf{X}_{k-1}$–$\mathbf{B}_k$, $\mathbf{X}_{k+1}$–$\mathbf{B}_k$–$\mathbf{B}_{k+1}$, $\mathbf{X}_{k-1}$–$\mathbf{B}_k$–$\mathbf{B}_{k+1}$ and $\mathbf{B}_k$–$\mathbf{X}_{k-1}$–$\mathbf{X}_{k+1}$.

The trade-off between the compression and preservation of relevant information is defined as the minimum achievable rate $I(\mathbf{X}_{k-1}; \mathbf{B}_k)$ subject to constraints on the information processing. We call it a *compact perception problem* which determines a trade-off between compression representation and predictive characteristics. Formally, this can be written as the following optimization.

$$\min_{p(\mathbf{B}_k|\mathbf{X}_{k-1}), p(\mathbf{B}_{k+1}|\mathbf{B}_k)} \quad I(\mathbf{X}_{k-1}; \mathbf{B}_k)$$

$$\text{subject to} \qquad \begin{aligned} I(\mathbf{X}_{k-1}; \mathbf{X}_k) - I(\mathbf{B}_k; \mathbf{X}_k) &\leq \epsilon_1 \\ I(\mathbf{B}_k; \mathbf{B}_{k+1}) - I(\mathbf{X}_{k-1}; \mathbf{X}_k) &\leq \epsilon_2 \\ I(\mathbf{X}_{k-1}; \mathbf{X}_{k+1}) - I(\mathbf{X}_{k+1}; \mathbf{B}_{k+1}) &\leq \epsilon_3 \end{aligned} \qquad (3)$$

where the constraints characterize the bounds on the desired prediction/compression at each step. We wish to lower bound (to guarantee) the prediction accuracies via lower bounding $I(\mathbf{B}_k; \mathbf{X}_k)$

and $I(\mathbf{X}_{k+1}; \mathbf{B}_{k+1})$, and upper bound (to limit) the compression level across hop via $I(\mathbf{B}_k; \mathbf{B}_{k+1})$. For example, $\epsilon_1$ bounds the accuracy of the prediction of $\mathbf{X}_k$ by $\mathbf{B}_k$. The information flow across alternate dynamical process is controlled by $\epsilon_2$. Lastly, $\epsilon_3$ defines closeness of prediction of $\mathbf{X}_{k+1}$ by $\mathbf{B}_{k+1}$. We show in the Section 3.1, and results in Section 4, that such trade-off can be alternatively studied by choice of the Lagrange parameters.

## 3 LEARNING INFORMATION PROCESSING FOR DYNAMICAL SYSTEMS

This section provides the main results concerning solving the compact perception optimization problem in equation (3) under the most general case. Next, this general result is used to study a high-dimensional continuous Gaussian distribution. Lastly, we describe an iterative method to update the corresponding parameters.

### 3.1 LEARNING INFORMATION PROPAGATION: SOLUTION

Finding the alternate dynamical representation, or $\mathbf{B}_k$ and $\mathbf{B}_{k+1}$, as stated in (3) requires to solve a variational problem. To solve this problem, we introduce the Lagrange multipliers $\beta$, $\lambda$ and $\gamma$ for the information processing constraints. Hence, we find the alternative representation (or IBH) by minimizing the following functional written using (3):

$$\mathcal{F}[p(\mathbf{B}_k|\mathbf{X}_{k-1}), p(\mathbf{B}_k), p(\mathbf{X}_k|\mathbf{B}_k), p(\mathbf{X}_{k+1}|\mathbf{B}_k), p(\mathbf{B}_{k+1}|\mathbf{B}_k), p(\mathbf{B}_{k+1})] =$$
$$I(\mathbf{X}_{k-1}; \mathbf{B}_k) - \beta I(\mathbf{B}_k; \mathbf{X}_k) + \lambda \left( I(\mathbf{B}_k; \mathbf{B}_{k+1}) - \gamma I(\mathbf{X}_{k+1}; \mathbf{B}_{k+1}) \right). \quad (4)$$

We can argue the following regarding the optimal solution which minimizes the equation (4).

**Theorem 1.** *The optimal solution that minimizes functional $\mathcal{F}$ in (4) satisfy the following self-consistent equations:*

$$
\begin{aligned}
p(\mathbf{B}_k|\mathbf{X}_{k-1}) \quad = \quad & \frac{p(\mathbf{B}_k)}{Z_1} \times \exp\left\{ -\beta D_{KL}\left(p(\mathbf{X}_k|\mathbf{X}_{k-1})||p(\mathbf{X}_k|\mathbf{B}_k)\right) \right. \\
& \left. - \lambda D_{KL}\left(p(\mathbf{B}_{k+1}|\mathbf{B}_k)||p(\mathbf{B}_{k+1})\right) \right. \\
& \left. - \lambda\gamma \mathbb{E}_{\mathbf{B}_{k+1}|\mathbf{B}_k}\left[ D_{KL}\left(p(\mathbf{X}_{k+1}|\mathbf{X}_{k-1})||p(\mathbf{X}_{k+1}|\mathbf{B}_{k+1})\right) \right] \right\}, \quad (5)
\end{aligned}
$$

$$
p(\mathbf{B}_{k+1}|\mathbf{B}_k) \quad = \quad \frac{p(\mathbf{B}_{k+1})}{Z_2} \exp\left\{ -\gamma D_{KL}\left(p(\mathbf{X}_{k+1}|\mathbf{B}_k)||p(\mathbf{X}_{k+1}|\mathbf{B}_{k+1})\right) \right\}, \quad (6)
$$

*where $Z_1$ and $Z_2$ are normalizing partition functions.*

The functional $\mathcal{F}$ in (4) may not be convex in the product space of the associated probability simplexes. Hence, it is difficult to obtain a global optimum, however, a stationary point (and locally optimal solution, in most of the cases) can be obtained using the following result.

**Corollary 1.** *The self-consistent equations in Theorem 1 can be used to write an iterative procedure to update the associated probabilities as equations (14)-(20).*

The iterative approach of Corollary 1 is detailed in the Appendix B. The idea is similar to the Blahut-Arimoto algorithm (Arimoto, 1972; Blahut, 1972), also observed in (Tishby et al., 2000). Finally, it remains to show the existence of a stationary point of the functional in (4). The convergence of the iterations in Corollary 1 is established through the following result.

**Lemma 1.** *The iterative procedure in Corollary 1 to minimize the functional $\mathcal{F}$ in (4) is convergent to a stationary point.*

The idea of proof (in Appendix C) is also somewhat similar to the EM McLachlan & Krishnan (1996) but with minimization of the functional. As in standard EM, in most of the cases, the stationary convergence point is local minimum (maximum in EM) of the functional.

**Corollary 2.** *The IBH solution in Theorem 1 reduces to IB in (2) upon setting $\mathbf{X}_{k-1} = \mathbf{X}_k$, and $\beta \to \infty$.*

*Proof.* With $\mathbf{X}_{k-1} = \mathbf{X}_k$, the functional of IBH in equation (4) can be written as

$$\mathcal{F} \quad = \quad (1 - \beta)I(\mathbf{X}_{k-1}; \mathbf{B}_k) + \lambda \left( I(\mathbf{B}_k; \mathbf{B}_{k+1}) - \gamma I(\mathbf{X}_{k+1}; \mathbf{B}_{k+1}) \right).$$

The functional $\mathcal{F}$ minimization in the limit of $\beta \to \infty$ has one solution of $p(\mathbf{B}_k|\mathbf{X}_{k-1}) = \mathbb{1}_{\mathbf{B}_k = \mathbf{X}_{k-1}}$ (such that $I(\mathbf{X}_{k-1}; \mathbf{B}_k) > 0$), and the problem reduces to minimize the following

$$\hat{\mathcal{F}} = I(\mathbf{X}_{k-1}; \mathbf{B}_{k+1}) - \gamma I(\mathbf{X}_{k+1}; \mathbf{B}_{k+1}),$$

where $\lambda$ can be dropped, as $\lambda \geq 0$. ∎

An advantage of using the optimization framework in (3) is that it can be generalized to any length of the given input dynamical process by properly repeating the second and third constraint. In this work, we have studied the case of length three of the input process. However, the same principles can be used to write the solution for any length $N$ of the dynamical system (the complete Figure 1), as will be presented in the future work.

### 3.2 MEASURING INFORMATION FLOW FOR LINEAR DYNAMICS

Many applications in machine learning involve time-series with multi-dimensional observations where each dimension is very likely to be correlated with others, and the state of observations evolves in time. Assuming that these observations are corrupted by Gaussian noise, then a linear dynamical system is a promising model for analyzing the provided time-series data. Thus, the system dynamics can be modeled similarly as the evolution of a stochastic time-invariant linear system. Finally, if we have Gaussian distributed initial-state of the linear dynamical system with additive Gaussian noise at each time step, then all future states are jointly Gaussian distributed. In this case, our framework learns the IBH through Gaussian random vectors.

Recall that the states $\mathbf{X}_k$ of the dynamical system under study are jointly Gaussian, and without loss of generality we assume that they are centered. As explained previously, we aim to design $\mathbf{B}_k$ and $\mathbf{B}_{k+1}$ to define an alternate representation which captures the dynamics of the relevant information. It is shown in the prior works of (Globerson & Tishby, 2004; Chechik et al., 2005) that for the problem setup of IB in which the input and output variables are jointly Gaussian, the optimum solution of the IB Lagrangian obtained by a stochastic transformation is also jointly Gaussian with the bottleneck's input. Consequently, $\mathbf{B}_k$ is jointly Gaussian with $\mathbf{X}_{k-1}$ and $\mathbf{B}_{k+1}$. Since RVs in consideration are mean centered and $\mathbf{X}_{k-1}, \mathbf{B}_k, \mathbf{B}_{k+1}$ are jointly Gaussian; the IBH variables can be very well represented as linear transformations of each other. Additionally, using the MC conditions from Section 2.2 we write the following linear relations.

$$\mathbf{B}_k = \mathbf{\Phi}\mathbf{X}_{k-1} + \boldsymbol{\xi}_k,$$
$$\mathbf{B}_{k+1} = \mathbf{\Delta}\mathbf{B}_k + \boldsymbol{\xi}_{k+1}, \tag{7}$$

where $\boldsymbol{\xi}_k$ and $\boldsymbol{\xi}_{k+1}$ are centered Gaussian random vectors independent of $\mathbf{X}_{k-1}$ and $\mathbf{B}_k$, respectively. Given the aforementioned settings, the solution of the minimization problem in equation (3) is determined by finding the matrices $\mathbf{\Phi}$ and $\mathbf{\Delta}$, and the covariance matrices $\mathbf{\Sigma}_{\boldsymbol{\xi}_k}$ and $\mathbf{\Sigma}_{\boldsymbol{\xi}_{k+1}}$. An iterative procedure using Corollary 1 to update the concerned parameters in (7) is presented as the following result.

**Theorem 2.** *Given the parameters $\beta$, $\lambda$ and $\gamma$, the Gaussian bottlenecks $\mathbf{B}_k = \mathbf{\Phi}\mathbf{X}_{k-1} + \boldsymbol{\xi}_k$ and $\mathbf{B}_{k+1} = \mathbf{\Delta}\mathbf{B}_k + \boldsymbol{\xi}_{k+1}$ are obtained by performing the following iterations over the parameters*

$$\mathbf{\Sigma}_{\boldsymbol{\xi}_k}^{(t+1)} = \left(\beta\,\mathbf{\Sigma}_{\mathbf{B}_k|\mathbf{X}_k}^{-1\,(t)} - (\beta-1)\,\mathbf{\Sigma}_{\mathbf{B}_k}^{-1\,(t)} + \lambda\,\mathbf{\Delta}^{T\,(t)}\mathbf{\Sigma}_{\mathbf{B}_{k+1}}^{-1\,(t)}\mathbf{\Delta}^{(t)}\right)^{-1},$$

$$\mathbf{\Phi}^{(t+1)} = \mathbf{\Sigma}_{\boldsymbol{\xi}_k}^{(t+1)}\left(\beta\,\mathbf{\Sigma}_{\mathbf{B}_k|\mathbf{X}_k}^{-1\,(t)}\mathbf{\Phi}^{(t)}(I - \mathbf{\Sigma}_{\mathbf{X}_{k-1}|\mathbf{X}_k}\mathbf{\Sigma}_{\mathbf{X}_{k-1}}^{-1})\right.$$
$$\left. + \lambda\gamma\,\mathbf{\Delta}^{T\,(t)}\mathbf{\Sigma}_{\mathbf{B}_{k+1}|\mathbf{X}_{k+1}}^{-1\,(t)}\mathbf{\Delta}^{(t)}\mathbf{\Phi}^{(t)}\mathbf{\Sigma}_{\mathbf{X}_{k+1}}^{-1}\mathbf{\Sigma}_{\mathbf{X}_{k+1}\mathbf{X}_{k-1}}\mathbf{\Sigma}_{\mathbf{X}_{k-1}}^{-1}\right), \tag{8}$$

$$\mathbf{\Sigma}_{\boldsymbol{\xi}_{k+1}}^{(t+1)} = \left(\gamma\mathbf{\Sigma}_{\mathbf{B}_{k+1}|\mathbf{X}_{k+1}}^{-1\,(t)} - (\gamma-1)\mathbf{\Sigma}_{\mathbf{B}_{k+1}}^{-1\,(t)}\right)^{-1},$$

$$\mathbf{\Delta}^{(t+1)} = \left(I - \frac{\gamma-1}{\gamma}\mathbf{\Sigma}_{\mathbf{B}_{k+1}|\mathbf{X}_{k+1}}^{(t)}\mathbf{\Sigma}_{\mathbf{B}_{k+1}}^{-1\,(t)}\right)^{-1}\mathbf{\Delta}^{(t)}\left(I - \mathbf{\Sigma}_{\mathbf{B}_k|\mathbf{X}_{k+1}}^{(t+1)}\mathbf{\Sigma}_{\mathbf{B}_k}^{-1\,(t+1)}\right),$$

*where $t$ is the iteration index.*

The detailed proof of the Theorem 2 is provided in the Appendix. In the next section, we show some numerical results generated using synthetic data.

## 4 SIMULATION EXPERIMENTS

We numerically evaluate the results of Theorem 1 and Theorem 2 using synthetically generated Gaussian distribution. Specifically, the covariance matrices for the input dynamical process

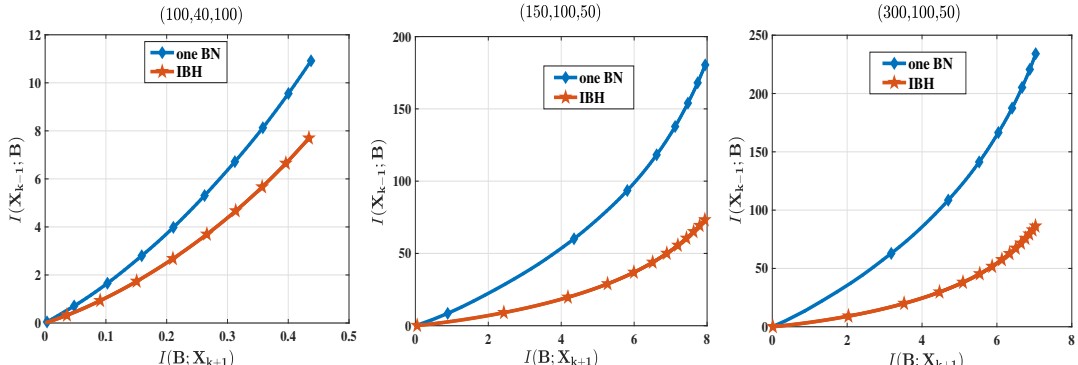

Figure 2: A comparison among single IB (designed locally) and IBH for various size tuples of $(\mathbf{X_{k-1}}, \mathbf{X_k}, \mathbf{X_{k+1}})$, respectively. The horizontal axis denotes required level of information transfer, while vertical is inversely proportional to compression levels.

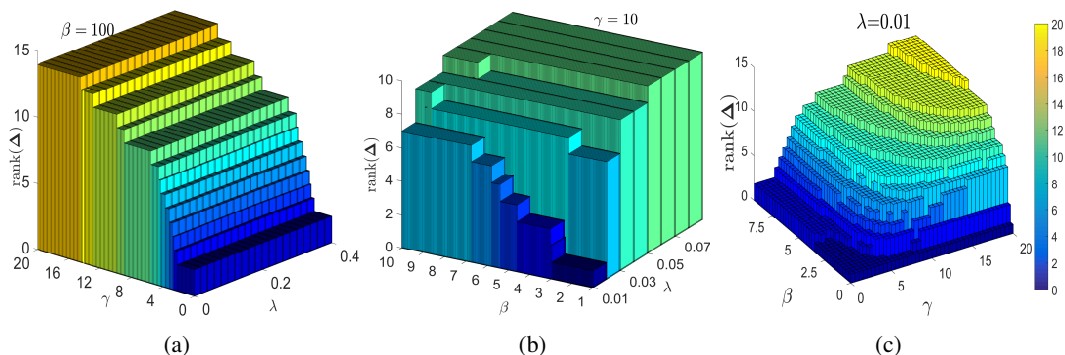

Figure 3: Variations of rank($\mathbf{\Delta}$) with Lagrange parameters $\beta, \lambda$ and $\gamma$, when dimension of input dynamics $(\mathbf{X}_{k-1}, \mathbf{X}_k, \mathbf{X}_{k+1})$ is $(40, 30, 20)$. The rank variation is presented by fixing one parameter in each scenario: $(\beta = 100, \lambda, \gamma)$ in (3a), $(\beta, \gamma = 10, \lambda)$ in (3b), and $(\beta, \gamma, \lambda = 0.01)$ in (3c).

$\mathbf{X}_{k-1}$–$\mathbf{X}_k$–$\mathbf{X}_{k+1}$ are generated numerically for a given size tuple, and we compute the parameters of (7) using (8). The quantities of interest are $I(\mathbf{B}_{k+1}; \mathbf{X}_{k+1})$ and $I(\mathbf{B}_{k+1}; \mathbf{X}_{k-1})$ which are indicators of prediction information and inverse of compression, respectively. We compare the prediction/compression behavior of the proposed approach of the alternate design of the dynamical system vs. designing local IB's between each hop. The local IB's are designed between two consecutive RVs in the dynamical system independently while the IBH is designed jointly. We show the distinction upon varying the dimensions of the input process $(\mathbf{X}_{k-1}, \mathbf{X}_k, \mathbf{X}_{k+1})$ in Figure 2. It is observed that the gap between prediction (for a fixed level of compression) grows with an increase in the input dimensions. The IBH by design takes into account the entire input dynamical system, and construct an alternate representation which provides better prediction at each step, by appropriate choice of the Lagrange parameters $\beta, \lambda, \gamma$.

The Lagrange parameters $(\beta, \lambda, \gamma)$ control the trade-off between compression and prediction at each step of the alternatively designed dynamical process. In the optimization problem (3), $\beta$ corresponds to the first constraint, and hence plays a deciding role in prediction accuracy of $\mathbf{B}_k$. The $\lambda$ corresponds to the second constraint, and will control the flow of relevant information across $\mathbf{B}_k$. Finally, $\lambda$ and $\gamma$ together tune the accuracy of prediction using $\mathbf{B}_{k+1}$. For example, in (7), the information tapping behavior can be visualized by inspecting the ranks of $\mathbf{\Phi}$ and $\mathbf{\Delta}$ matrices upon varying $(\beta, \lambda, \gamma)$. In Figure 3a, the rank($\mathbf{\Delta}$) increases upon increasing $\gamma$ for each choice of $\lambda$. Since $\lambda$ appears in front of both prediction and compression expression in (4), it has little effect on the rank($\mathbf{\Delta}$) for a fixed $\gamma$ and $\beta$. Next, in Figure 3b, we observe dynamical effects of the information flow. By fixing $\lambda$, we limit the information acceptance of $\mathbf{B}_{k+1}$, hence the parameter $\beta$ can only increase the rank($\mathbf{\Delta}$) up to a certain limit by allowing maximum information through $\mathbf{B}_k$. We witness in Figure 3b, that with higher $\lambda$, the parameter $\beta$ quickly increase the rank($\mathbf{\Delta}$). Now, for a fixed $\lambda$, both

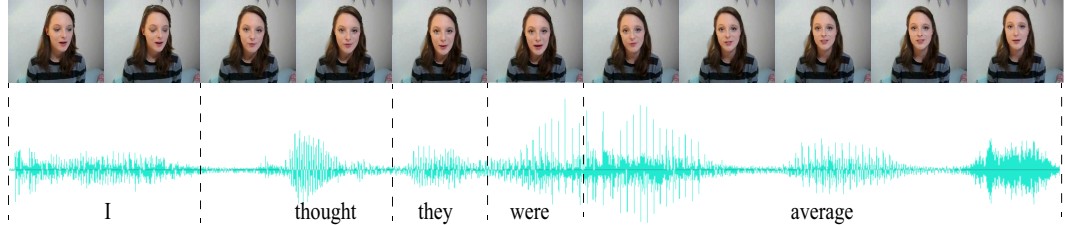

Figure 4: The alignment of three modalities (text, visual and audio) with averaging of visual and audio features corresponding to the time boundaries obtained from text.

$\beta$ and $\gamma$ intertwine with each other to decide rank$(\mathbf{\Delta})$. By fixing $\beta$, we limit the input information through $\mathbf{B}_k$ to $\mathbf{B}_{k+1}$, or availability, and hence $\gamma$ can only increase the rank$(\mathbf{\Delta})$ up to certain extent. Similarly, fixing $\gamma$ limits the maximum information that $\mathbf{B}_{k+1}$ can process, and therefore $\beta$ can do best narrowly up to some extent. We witness this hyperbolic behavior in Figure 3c.

## 5 REAL-WORLD DATA: MULTIMODAL SENTIMENT INTENSITY

In this section, we apply the ideas of IBH to extract the features from the challenging multimodal datasets available in the form of time-series. Particularly, we have used the CMU Multimodal Sentiment Analysis (CMU-MOSI) dataset (Zadeh et al., 2016) which consists of a total of 2198 videos. Each video comprises one speaker expressing their opinion in front of the camera. The available modalities from the dataset are text, visual and audio. The goal of the dataset is to perform a discriminative task of predicting the speaker sentiment using the available modalities. From the three modalities of text, visual and audio, the corresponding features are extracted using GloVe word embeddings (Pennington et al., 2014), Facet (iMotions, 2017) and COVAREP (Degottex et al., 2014), respectively. The extracted feature size are 300 for text, 74 for audio, and 46 for visual component.

The extracted features in the form of time-series are aligned across modalities as shown in Figure 4. Specifically, multiple features of visual and audio modality (due to high frame/sampling rate) are time averaged with boundaries corresponding to the text component. For each speaker, we have a maximum of 20 words with three modalities and the corresponding sentiment intensity being a real number ranging from $-3$ to 3, with negative values representing negative sentiments and vice-versa.

Some of the prior work in the multimodal representation learning include Discriminative representation learning (Zadeh et al., 2018a; 2017; 2018b; Chaplot et al., 2017) and Generative representation learning (Sohn et al., 2014; Srivastava & Salakhutdinov, 2014; Suzuki et al., 2016). Interestingly, recent work Tsai et al. (2018) proposed Multimodal Factorized Model (MFM) that exploits the fusion of both these techniques. The work factorizes the data into discriminative factors and modality-specific generative factors. We note that, at the very core, the challenge in learning patterns from multiple modalities is to address the *complex inter and well as intra dependencies across them*. Since IBH is capable of compressing the given dynamical model into an alternate version stochastically, we propose to, (i) first map the multiple modalities into a time-varying linear dynamical system; and then (ii) use IBH to identify the complex inter and well as intra dependencies across them in a reduced dimensional model for better discrimination using simple machine learning classifiers.

The IBH is applied to various modalities to capture the information flow patterns across them and in the compressed fashion. We have taken three modalities to be in the following Markov Chain, text–audio–visual. The intuitive reason behind this assumption is that *text* is the most informative modality for sentiment and hence the first state, while *audio* and *visual* follows *text* in the Markov chain assuming the speaker is being honest in speaking and making visual expressions for a particular sentiment. The data is mean centered and the covariances of three modalities are estimated for each speaker in the training as well as the testing dataset. However, we have only 20 words and hence 20 samples to estimate the covariance matrix of much larger dimensions. To remedy this fewer samples problem, we have resorted to something called pooling of the covariance matrix as follows. The covariance matrix of any modality for the $i$th speaker is written as $\mathbf{\Sigma}_i = \alpha\hat{\mathbf{\Sigma}}_i + (1-\alpha)\mathbf{\Sigma}_{\text{pool}}$, where $\hat{\mathbf{\Sigma}}_i$ is the estimated covariance from 20 samples, and $\mathbf{\Sigma}_{\text{pool}}$ is called pooled covariance matrix

| Feature Selection | # features | binary | 7-class | MAE |
|---|---|---|---|---|
| Early Fusion | 8400 | 51.9% | 19.1% | 1.382 |
| MFM (Tsai et al., 2018) | 8400 | 77.3% | 35.4% | 0.961 |
| IBH | 1150 | 77.4% | 39.4% | 0.922 |

Table 1: Comparison of sentiment prediction with three different methods of evaluations for early fusion of features vs features extraction by IBH for CMU-MOSI.

which is estimated by taking all of the training data. The parameter $\alpha$ is used to make a trade-off between these two matrices and is usually chosen close to 1. The covariance matrices are fed to the algorithm in Theorem 2 to estimate $\Phi$ and $\Delta$ matrices. The $\Phi$ matrix has components for interactions across text and audio, while $\Delta$ has entries representing interactions with the compressed version of text-audio inter-dependencies as well as with the visual modality. Therefore, the entries of $\Delta$ matrix are good candidates for representing inter as well as intra dependencies across all three modalities. Hence, we use $\Delta$ matrix as a feature to predict the sentiment intensity, as this matrix is central to information propagation from the text, through audio, to visual component. The low rank of $\Delta$ will be key in reducing the number of features. The values of parameters $(\beta, \lambda, \gamma)$ are chosen such that maximum information flow to first bottleneck, by setting high value for $\beta$, and $\lambda$ close to 1, and low values for $\gamma$ for better compression.

The results are reported for various evaluation methods, namely binary (with the positive and negative sentiment) and $7-$ class classification, and Mean Average Error (MAE) for regression. We compare our performance with naive early fusion of features in raw format, most recent results in the multimodal neural networks (Tsai et al., 2018) vs. features processed by IBH, and the numerical results are presented in Table 1. Support Vector Machines (SVM) is used in the cases of early fusion and IBH. The processing of features by IBH has a two-fold advantage: First, due to compression, the dimensionality of the input can be reduced from 8400 to 1150. Second, the performance with respect to various metrics is better.

## 6 CONCLUSION

In this paper, we have introduced a novel information-theoretic inspired approach to learn the compact dynamics of a time-varying complex system. The trade-off between the predictive accuracy and the compactness of the mathematical representation is formulated as a multi-hop compact perception optimization problem. A key ingredient to solve the aforementioned problem is to exploit variational calculus in order to derive the general solution expressions. Additionally, we have investigated the guaranteed convergence of the proposed iterative algorithm. Moreover, considering a specific class of distributions (Gaussian), we have provided closed-form expressions for the model parameters' update in our algorithm. Interestingly, the proposed compact perception shows improvements in prediction with reduced dimension on challenging real-world problems.

The quantification of information flow across a dynamical system can have an enormous impact on understanding and improving the current state-of-the-art in neural networks as realized in (Tishby & Zaslavsky, 2015). Moreover, modeling with dynamical systems is a standard approach, and by using the proposed framework, we can make a better compact representation of the system. The driving force of a dynamical system can enforce different behaviors of information flow, as realized in defining dynamical entropy by (Sinai, 1959). Therefore, measuring the information flow can help in estimating/differentiating the actual driving component behind the observed activities. Such concepts are useful in predicting brain imagined tasks from observed electroencephalogram activities.

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

The appendix is arranged as follows: In the Section A, we provide the proof of Theorem 1. In the Section B, we provide the iterative procedure (mentioned as Corollary 1) to minimize the functional in (4). Next, in Section C, we present the detailed proof of the Lemma 1, and finally, in Section D, a detailed proof of Theorem 2 is presented.

## A    PROOF OF THEOREM 1

*Proof.* For the sake of simplicity, a sketch of the proof is given for discrete variables. The Lagrangian associated with the minimization problem is the following

$$\mathcal{L} = \mathcal{F} - \sum_{\mathbf{B}_k, \mathbf{X}_{k-1}} \alpha_1(\mathbf{X}_{k-1}) p(\mathbf{B}_k | \mathbf{X}_{k-1}) - \sum_{\mathbf{B}_k, \mathbf{B}_{k+1}} \alpha_2(\mathbf{B}_k) p(\mathbf{B}_{k+1} | \mathbf{B}_k), \tag{9}$$

where $\alpha_1(\mathbf{X}_{k-1})$ and $\alpha_2(\mathbf{B}_k)$ are Lagrange multipliers for the normalization of the distributions $p(\mathbf{B_k} | \mathbf{X_{k-1}})$ and $p(\mathbf{B}_{k+1} | \mathbf{B}_k)$, respectively. Taking the derivative of each term of the Lagrangian $\mathcal{L}$ with respect to $p(\mathbf{B}_k | \mathbf{X}_{k-1})$, we have

$$\frac{\delta I(\mathbf{X}_{k-1}; \mathbf{B}_k)}{\delta p(\mathbf{B}_k | \mathbf{X}_{k-1})} = p(\mathbf{X}_{k-1}) \log \frac{p(\mathbf{B}_k | \mathbf{X}_{k-1})}{p(\mathbf{B}_k)},$$

$$\frac{\delta I(\mathbf{X}_k; \mathbf{B}_k)}{\delta p(\mathbf{B}_k | \mathbf{X}_{k-1})} = p(\mathbf{X}_{k-1}) \sum_{\mathbf{X}_k} p(\mathbf{X}_k | \mathbf{X}_{k-1}) \log \frac{p(\mathbf{X}_k | \mathbf{B}_k)}{p(\mathbf{X}_k)},$$

$$\frac{\delta I(\mathbf{B}_k; \mathbf{B}_{k+1})}{\delta p(\mathbf{B}_k | \mathbf{X}_{k-1})} = p(\mathbf{X}_{k-1}) D_{KL}(p(\mathbf{B}_{k+1} | \mathbf{B}_k) || p(\mathbf{B}_{k+1})),$$

$$\frac{\delta I(\mathbf{X}_{k+1}; \mathbf{B}_{k+1})}{\delta p(\mathbf{B}_k | \mathbf{X}_{k-1})} = -p(\mathbf{X}_{k-1}) \sum_{\mathbf{B}_{k+1}} p(\mathbf{B}_{k+1} | \mathbf{B}_k) D_{KL}(p(\mathbf{X}_{k+1} | \mathbf{X}_{k-1}) || p(\mathbf{X}_{k+1} | \mathbf{B}_{k+1}))$$
$$+ p(\mathbf{X}_{k-1}) D_{KL}(p(\mathbf{X}_{k+1} | \mathbf{X}_{k-1}) || p(\mathbf{X}_{k+1})). \tag{10}$$

Setting the derivative of the Lagrangian equal to zero and arranging the terms we obtain the self consistent equation (5). Note that all the constant terms in the derivative independent of $\mathbf{B}_k$ will be captured by the Lagrange multiplier $\alpha_1(\mathbf{X}_{k-1})$. The derivative of the Lagrangian $\mathcal{L}$ with respect to $p(\mathbf{B}_{k+1} | \mathbf{B}_k)$ involves only the two last terms from the functional $\mathcal{F}$ and the term that ensures the normalization condition. Then, we have

$$\frac{\delta I(\mathbf{B}_k; \mathbf{B}_{k+1})}{\delta p(\mathbf{B}_{k+1} | \mathbf{B}_k)} = p(\mathbf{B}_k) \log \frac{p(\mathbf{B}_{k+1} | \mathbf{B}_k)}{p(\mathbf{B}_{k+1})}, \tag{11}$$

$$\frac{\delta I(\mathbf{X}_{k+1}; \mathbf{B}_{k+1})}{\delta p(\mathbf{B}_{k+1} | \mathbf{B}_k)} = p(\mathbf{B}_k) \sum_{\mathbf{X}_{k+1}} p(\mathbf{X}_{k+1} | \mathbf{B}_k) \log \frac{p(\mathbf{X}_{k+1} | \mathbf{B}_{k+1})}{p(\mathbf{X}_{k+1})}. \tag{12}$$

Thus, the variational condition is written as follows

$$p(\mathbf{B_k}) \left( \log \frac{p(\mathbf{B}_{k+1} | \mathbf{B}_k)}{p(\mathbf{B}_{k+1})} + \gamma D_{KL}(p(\mathbf{X}_{k+1} | \mathbf{B}_k) || p(\mathbf{X}_{k+1} | \mathbf{B}_{k+1})) - \overline{\alpha}_2(\mathbf{B}_k) \right) = 0, \tag{13}$$

where $\overline{\alpha}_2(\mathbf{B}_k)$ is the summation of the Lagrange multiplier $\alpha_2(\mathbf{B}_k)$ and the terms independent of $\mathbf{B}_{k+1}$, and hence the equation (6) follows. ∎

## B    ITERATIVE SOLUTION IN COROLLARY 1

The self-consistent equations derived in Theorem 1 to minimize the functional $\mathcal{F}$ in (4) can be used to write the following set of iterative equations.

$$p^{(t+1)}(\mathbf{B}_k | \mathbf{X}_{k-1}) = \frac{p^{(t)}(\mathbf{B}_k)}{Z_1^{(t+1)}(\beta, \lambda, \gamma, \mathbf{X}_{k-1})} \times \exp \left\{ -\beta D_{KL} \left( p(\mathbf{X}_k | \mathbf{X}_{k-1}) || p^{(t)}(\mathbf{X}_k | \mathbf{B}_k) \right) \right.$$
$$- \lambda D_{KL} \left( p^{(t)}(\mathbf{B}_{k+1} | \mathbf{B}_k) || p^{(t)}(\mathbf{B}_{k+1}) \right)$$
$$\left. - \lambda \gamma \mathbb{E}_{\mathbf{B}_{k+1} | \mathbf{B}_k}^{(t)} \left[ D_{KL} \left( p(\mathbf{X}_{k+1} | \mathbf{X}_{k-1}) || p^{(t)}(\mathbf{X}_{k+1} | \mathbf{B}_{k+1}) \right) \right] \right\}, \tag{14}$$

$$p^{(t+1)}(\mathbf{B}_k) = \sum_{\mathbf{X}_{k-1}} p^{(t+1)}(\mathbf{B}_k|\mathbf{X}_{k-1})\, p(\mathbf{X}_{k-1}), \tag{15}$$

$$p^{(t+1)}(\mathbf{X}_k|\mathbf{B}_k) = \sum_{\mathbf{X}_{k-1}} p(\mathbf{X}_k|\mathbf{X}_{k-1})\, p^{(t+1)}(\mathbf{X}_{k-1}|\mathbf{B}_k), \tag{16}$$

$$p^{(t+1)}(\mathbf{X}_{k+1}|\mathbf{B}_k) = \sum_{\mathbf{X}_k} p(\mathbf{X}_{k+1}|\mathbf{X}_k)\, p^{(t+1)}(\mathbf{X}_k|\mathbf{B}_k), \tag{17}$$

$$p^{(t+1)}(\mathbf{B}_{k+1}|\mathbf{B}_k) = \frac{p^{(t)}(\mathbf{B}_{k+1})}{Z_2^{(t+1)}(\gamma,\mathbf{B}_k)} \exp\left\{-\gamma D_{KL}\left(p^{(t+1)}(\mathbf{X}_{k+1}|\mathbf{B}_k)||p^{(t)}(\mathbf{X}_{k+1}|\mathbf{B}_{k+1})\right)\right\}, \tag{18}$$

$$p^{(t+1)}(\mathbf{B}_{k+1}) = \sum_{\mathbf{B}_k} p^{(t+1)}(\mathbf{B}_{k+1}|\mathbf{B}_k)\, p^{(t+1)}(\mathbf{B}_k), \tag{19}$$

$$p^{(t+1)}(\mathbf{B}_{k+1}|\mathbf{X}_{k+1}) = \sum_{\mathbf{B}_k,\mathbf{X}_{k-1}} p^{(t+1)}(\mathbf{B}_{k+1}|\mathbf{B}_k)\, p^{(t+1)}(\mathbf{B}_k|\mathbf{X}_{k-1})\, p(\mathbf{X}_{k-1}|\mathbf{X}_{k+1}), \tag{20}$$

where in equations (15)-(17), (19) and (20) we have used the Markov assumption as stated in Section 2.2.

## C    PROOF OF LEMMA 1

*Proof.* The proof of the Lemma 1 can be divided into two parts. First, we show that the $\mathcal{F}$ in (4) is lower-bounded. Next, we show that each iteration using the equations (14)-(20) monotonically decrease the functional.

**Lower bound:** Let us consider the following alternate functional.

$$\tilde{\mathcal{F}} = I(\mathbf{X}_{k-1};\mathbf{B}_k) + \beta\mathbb{E}_{\mathbf{X}_{k-1},\mathbf{B}_k} D_{KL}(p(\mathbf{X}_k|\mathbf{X}_{k-1})||p(\mathbf{X}_k|\mathbf{B}_k)) + \lambda I(\mathbf{B}_k;\mathbf{B}_{k+1})$$
$$\lambda\gamma\,\mathbb{E}_{\mathbf{X}_{k-1},\mathbf{B}_k}\mathbb{E}_{\mathbf{B}_{k+1}|\mathbf{B}_k} D_{KL}(p(\mathbf{X}_{k+1}|\mathbf{X}_{k-1})||p(\mathbf{X}_{k+1}|\mathbf{B}_{k+1})). \tag{21}$$

It can be readily verified that the functional $\tilde{\mathcal{F}} \geq 0$ for given non-negative constants $\beta, \lambda$ and $\gamma$. Also, the $\tilde{\mathcal{F}}$ in (21) can be expanded as

$$\tilde{\mathcal{F}} = I(\mathbf{X}_{k-1};\mathbf{B}_k) + \beta\mathbb{E}_{\mathbf{X}_{k-1},\mathbf{B}_k} \sum_{\mathbf{X}_k} p(\mathbf{X}_k|\mathbf{X}_{k-1})\log\frac{p(\mathbf{X}_k|\mathbf{X}_{k-1})}{p(\mathbf{X}_k|\mathbf{B}_k)}$$

$$+ \lambda I(\mathbf{B}_k;\mathbf{B}_{k+1}) + \lambda\gamma\,\mathbb{E}_{\mathbf{X}_{k-1},\mathbf{B}_k} \sum_{\mathbf{B}_{k+1},\mathbf{X}_{k+1}} p(\mathbf{X}_{k+1}|\mathbf{X}_{k-1})\log\frac{p(\mathbf{X}_{k+1}|\mathbf{X}_{k-1})}{p(\mathbf{X}_{k+1}|\mathbf{B}_{k+1})}$$

$$= I(\mathbf{X}_{k-1};\mathbf{B}_k) + \beta(I(\mathbf{X}_{k-1};\mathbf{X}_k) - I(\mathbf{X}_k;\mathbf{B}_k)) + \lambda I(\mathbf{B}_k;\mathbf{B}_{k+1})$$
$$+ \lambda\gamma(I(\mathbf{X}_{k-1};\mathbf{X}_{k+1}) - I(\mathbf{X}_{k+1};\mathbf{B}_{k+1})). \tag{22}$$

Since the functional $\tilde{\mathcal{F}}$ in the equation (22) differs from the functional $\mathcal{F}$ in (4) only in constants, therefore, $\mathcal{F}$ is lower bounded as well.

**Monotonicity:** For proving monotonic decrement of the functional $\mathcal{F}$, we will use the formulation similar to (Arimoto, 1972). First, let us consider the following observation made in (Tishby et al., 2000).

For a given joint distribution $p(X,Y)$, we can write the following.

$$p(Y) = \arg\min_{\phi(Y)} \mathbb{E}_X D_{KL}(p(Y|X)||\phi(Y)), \tag{23}$$

where the minimization is performed over the probability simplex of $\phi(Y)$ such that the joint distribution is $p(X,Y)$. The functional $\tilde{\mathcal{F}}$ in (22) can be written in its most general form as

$$\tilde{\mathcal{F}}(p_1,p_2,\phi_1,\phi_2) = \mathbb{E}_{\mathbf{X}_{k-1},\mathbf{B}_k|\mathbf{X}_{k-1}\sim p_1}\log\frac{p_1}{p(\mathbf{B}_k)}$$
$$+ \beta\mathbb{E}_{\mathbf{X}_{k-1},\mathbf{B}_k|\mathbf{X}_{k-1}\sim p_1} D_{KL}(p(\mathbf{X}_k|\mathbf{X}_{k-1})||\phi_1)$$

$$+ \lambda \mathbb{E}_{\mathbf{X}_{k-1}, \mathbf{B}_k | \mathbf{X}_{k-1} \sim p_1} \mathbb{E}_{\mathbf{B}_{k+1} | \mathbf{B}_k \sim p_2} \log \frac{p_2}{p(\mathbf{B}_k)}$$

$$+ \lambda \gamma \mathbb{E}_{\mathbf{X}_{k-1}, \mathbf{B}_k | \mathbf{X}_{k-1} \sim p_1} \mathbb{E}_{\mathbf{B}_{k+1} | \mathbf{B}_k \sim p_2} D_{KL}(p(\mathbf{X}_{k+1} | \mathbf{X}_{k-1}) || \phi_2), \quad (24)$$

where $p_1 = p(\mathbf{B}_k | \mathbf{X}_{k-1}), p_2 = p(\mathbf{B}_{k+1} | \mathbf{B}_k), \phi_1 = \phi_1(\mathbf{X}_k | \mathbf{B}_k)$ and $\phi_2 = \phi_2(\mathbf{X}_{k+1} | \mathbf{B}_{k+1})$, therefore, $\tilde{\mathcal{F}}$ reduces to the form in (22) upon setting $\phi_1 = p(\mathbf{X}_k | \mathbf{B}_k)$ and $\phi_2 = p(\mathbf{X}_{k+1} | \mathbf{B}_{k+1})$. With the objective of minimizing the functional $\tilde{\mathcal{F}}$, we can write its value at iteration $t$ as $\tilde{\mathcal{F}}^{(t)} = \hat{\mathcal{F}}(p_1^{(t)}, p_2^{(t)}, \phi_1^{(t)}, \phi_2^{(t)})$. The iterations to minimize $\tilde{\mathcal{F}}$ will involve the successive choice of tuple $(p_1, p_2, \phi_1, \phi_2)$. At iteration $t$, let us assume that we have chosen $p_1^{(t)}$ and $p_2^{(t)}$, and we define the following

$$\begin{aligned} \mathcal{G}(t,t) &= \min_{\phi_1, \phi_2} \tilde{\mathcal{F}}(p_1^{(t)}, p_2^{(t)}, \phi_1, \phi_2) \\ &= \tilde{\mathcal{F}}(p_1^{(t)}, p_2^{(t)}, \phi_1^{(t)}, \phi_2^{(t)}). \end{aligned} \quad (25)$$

Using equation (23), it easily follows that $\phi_1^{(t)} = p^{(t)}(\mathbf{X}_k | \mathbf{B}_k)$ and $\phi_2^{(t)} = p^{(t)}(\mathbf{X}_{k+1} | \mathbf{B}_{k+1})$, and hence $\mathcal{G}(t,t) = \tilde{\mathcal{F}}(t)$. Now, for fixed $p_2^{(t)}, \phi_1^{(t)}, \phi_2^{(t)}$, it can be easily realized that the $\tilde{\mathcal{F}}$ is convex in $p_1$, therefore, minimizing $\tilde{\mathcal{F}}$ with respect to (w.r.t.) $p_1$ will involve writing Lagrangian, and then differentiation, and setting to zero. This step is similar to the Theorem 1, and we can write that

$$p_1^{(t+1)} = \arg \min_{p_1} \tilde{\mathcal{F}}(p_1, p_2^{(t)}, \phi_1^{(t)}, \phi_2^{(t)}), \quad (26)$$

where the resulting solution is (5), and hence $p_1^{(t+1)}$ will have the expression as in (14). Similarly, for fixed $p_1^{(t+1)}, \phi_1^{(t)}, \phi_2^{(t)}$, the $\tilde{\mathcal{F}}$ is convex in $p_2$, and the same steps follow to obtain $p_2^{(t+1)}$ as written in (18). It should be noted that the choice to perform minimization w.r.t. $p_1$ before $p_2$ is arbitrary, and the reverse can also be performed. This will change the order of iteration index in equations (14)-(20) accordingly. Using equations (25) and (26), we can conclude that

$$\tilde{\mathcal{F}}^{(t+1)} = \mathcal{G}(t+1, t+1) \le \mathcal{G}(t+1, t) \le \mathcal{G}(t, t) = \tilde{\mathcal{F}}^{(t)},$$

and therefore, iterating equations (14)-(20) written using the self-consistent equations of Theorem 1 minimizes $\tilde{\mathcal{F}}^{(t)}$ monotonically. Since $\tilde{\mathcal{F}}$ and $\mathcal{F}$ differs only in constant, it reduces $\mathcal{F}$ as well monotonically from above. ∎

## D  PROOF OF THEOREM 2

*Proof.* For a multivariate random variable $\mathbf{X}$, $\mathbf{X} \in \mathbb{R}^n$ with Gaussian distribution, i.e. $\mathbf{X} \sim \mathcal{N}(\boldsymbol{\mu}, \boldsymbol{\Sigma})$, the entropy can be written as

$$H(\mathbf{X}) = \frac{1}{2} \log \det(\boldsymbol{\Sigma}) + c, \quad (27)$$

where $c$ is constant for a given dimension $n$. Using equation (27), the KL-divergence between two Gaussian distributed random variables, $\mathbf{X}_1 \sim \mathcal{N}(\boldsymbol{\mu}_1, \boldsymbol{\Sigma}_1)$ and $\mathbf{X}_2 \sim \mathcal{N}(\boldsymbol{\mu}_2, \boldsymbol{\Sigma}_2)$ of the same dimensions, is written as

$$D_{KL}(p(\mathbf{X}_1) || p(\mathbf{X}_2)) = \frac{1}{2} \log \frac{\det(\boldsymbol{\Sigma}_2)}{\det(\boldsymbol{\Sigma}_1)} + \frac{1}{2} \text{tr}(\boldsymbol{\Sigma}_2^{-1} \boldsymbol{\Sigma}_1) + \frac{1}{2} (\boldsymbol{\mu}_2 - \boldsymbol{\mu}_1)^T \boldsymbol{\Sigma}_2^{-1} (\boldsymbol{\mu}_2 - \boldsymbol{\mu}_1). \quad (28)$$

We have assumed that the given data is centered, hence all considered random variables will have zero mean, i.e., for characterizing each random variable, we only need the corresponding covariance matrix. Let us revisit the linear transformation model for the IBH.

$$\begin{aligned} \mathbf{B}_k &= \boldsymbol{\Phi} \mathbf{X}_{k-1} + \boldsymbol{\xi}_k, \\ \mathbf{B}_{k+1} &= \boldsymbol{\Delta} \mathbf{B}_k + \boldsymbol{\xi}_{k+1}. \end{aligned}$$

Now, to completely specifying the model, we have to determine the constant matrices $\boldsymbol{\Phi}, \boldsymbol{\Delta}$ and the covariances of $\boldsymbol{\xi}_k, \boldsymbol{\xi}_{k+1}$. Since entropy is well defined for Gaussian distribution, as in (27), due to their nice tail distribution, we can use Theorem 1 and equation (28) to write the self-consistent

transition probabilities. First, expressions of the necessary KL-divergence terms are expanded as follows.

$$D_{KL}(p(\mathbf{X}_k|\mathbf{X}_{-1})||p(\mathbf{X}_k|\mathbf{B}_k)) = c + \frac{1}{2}(\mathbb{E}\,\mathbf{X}_k|\mathbf{X}_{-1} - \mathbb{E}\,\mathbf{X}_k|\mathbf{B}_k)^T \mathbf{\Sigma}_{\mathbf{X}_k|\mathbf{B}_k}^{-1}$$
$$(\mathbb{E}\,\mathbf{X}_k|\mathbf{X}_{-1} - \mathbb{E}\,\mathbf{X}_k|\mathbf{B}_k), \qquad (29)$$

$$D_{KL}(p(\mathbf{B}_{k+1}|\mathbf{B}_k)||p(\mathbf{B}_{k+1})) = c + \frac{1}{2}(\mathbb{E}\,\mathbf{B}_{k+1}|\mathbf{B}_k)^T \Sigma_{\mathbf{B_{k+1}}}^{-1} \mathbb{E}\,\mathbf{B}_{k+1}|\mathbf{B}_k, \qquad (30)$$

$$D_{KL}(p(\mathbf{X}_{k+1}|\mathbf{X}_{-1})||p(\mathbf{X}_{k+1}|\mathbf{B}_{k+1})) = c + \frac{1}{2}(\mathbb{E}\,\mathbf{X}_{k+1}|\mathbf{X}_{-1} - \mathbb{E}\,\mathbf{X}_{k+1}|\mathbf{B}_{k+1})^T \mathbf{\Sigma}_{\mathbf{X}_{k+1}|\mathbf{B}_{k+1}}^{-1}$$
$$(\mathbb{E}\,\mathbf{X}_{k+1}|\mathbf{X}_{-1} - \mathbb{E}\,\mathbf{X}_{k+1}|\mathbf{B}_{k+1}), \qquad (31)$$

where $c$ are different constants at each step and,

$$\begin{aligned}
\mathbb{E}\,\mathbf{X}_k|\mathbf{B}_k &= \mathbf{\Sigma}_{\mathbf{X}_k\mathbf{B}_k}\mathbf{\Sigma}_{\mathbf{B}_k}^{-1}\mathbf{B}_k = \mathbf{\Xi}_k\mathbf{B}_k, \\
\mathbf{\Sigma}_{\mathbf{X}_k|\mathbf{B}_k} &= \mathbf{\Sigma}_{\mathbf{X}_k} - \mathbf{\Sigma}_{\mathbf{X}_k\mathbf{B}_k}\mathbf{\Sigma}_{\mathbf{B}_k}^{-1}\mathbf{\Sigma}_{\mathbf{B}_k\mathbf{X}_k} \\
&= \mathbf{\Sigma}_{\mathbf{X}_k} - \mathbf{\Sigma}_{\mathbf{X}_k\mathbf{X}_{k-1}}\mathbf{\Phi}^T\mathbf{\Sigma}_{\mathbf{B}_k}^{-1}\mathbf{\Phi}\mathbf{\Sigma}_{\mathbf{X}_{k-1}\mathbf{X}_k}, \\
\mathbb{E}\,\mathbf{B}_{k+1}|\mathbf{B}_k &= \mathbf{\Delta}\mathbf{B}_k, \\
\mathbf{\Sigma}_{\mathbf{B}_{k+1}|\mathbf{B}_k} &= \mathbf{\Sigma}_{\boldsymbol{\xi}_{k+1}}, \\
\mathbb{E}\,\mathbf{X}_{k+1}|\mathbf{B}_{k+1} &= \mathbf{\Sigma}_{\mathbf{X}_{k+1}\mathbf{X}_{k-1}}\mathbf{\Phi}^T\mathbf{\Delta}^T\mathbf{\Sigma}_{\mathbf{B}_{k+1}}^{-1}\mathbf{B}_{k+1} = \mathbf{\Theta}_{k+1}\mathbf{B}_{k+1}, \\
\mathbf{\Sigma}_{\mathbf{X}_{k+1}|\mathbf{B}_{k+1}} &= \mathbf{\Sigma}_{\mathbf{X}_{k+1}} - \mathbf{\Sigma}_{\mathbf{X}_{k+1}\mathbf{X}_{k-1}}\mathbf{\Phi}^T\mathbf{\Delta}^T\mathbf{\Sigma}_{\mathbf{B}_{k+1}}^{-1}\mathbf{\Delta}\mathbf{\Phi}\mathbf{\Sigma}_{\mathbf{X}_{k-1}\mathbf{X}_{k+1}}.
\end{aligned}$$

Also, using (31), we can write that

$$\begin{aligned}
\mathbb{E}_{\mathbf{B}_{k+1}|\mathbf{B}_k}\,D_{KL}\left(p(\mathbf{X}_{k+1}|\mathbf{X}_{-1})||p(\mathbf{X}_{k+1}|\mathbf{B}_{k+1})\right) &= c \\
&- \mathbb{E}_{\mathbf{B}_{k+1}|\mathbf{B}_k}\,\mathbf{B}_{k+1}^T\mathbf{\Theta}_{k+1}^T\mathbf{\Sigma}_{\mathbf{X}_{k+1}|\mathbf{B}_{k+1}}^{-1}\mathbb{E}\,\mathbf{X}_{k+1}|\mathbf{X}_{-1} \\
&+ \frac{1}{2}\mathbb{E}_{\mathbf{B}_{k+1}|\mathbf{B}_k}\,\mathbf{B}_{k+1}^T\mathbf{\Theta}_{k+1}^T\mathbf{\Sigma}_{\mathbf{X}_{k+1}|\mathbf{B}_{k+1}}^{-1}\mathbf{\Theta}_{k+1}\mathbf{B}_{k+1} \\
&= c' - \mathbf{B}_k^T\mathbf{\Delta}^T\mathbf{\Theta}_{k+1}^T\mathbf{\Sigma}_{\mathbf{X}_{k+1}|\mathbf{B}_{k+1}}^{-1}\mathbb{E}\,\mathbf{X}_{k+1}|\mathbf{X}_{-1}. \qquad (32)
\end{aligned}$$

Upon substituting the equations (29)-(31) and (32) in Theorem 1, we obtain the following self-consistent equations.

$$\begin{aligned}
\log p(\mathbf{B}_k|\mathbf{X}_{-1}) &= c - \frac{1}{2}\mathbf{B}_k^T\mathbf{\Sigma}_{\mathbf{B}_k}^{-1}\mathbf{B}_k - \frac{\beta}{2}\mathbf{B}_k^T\mathbf{\Xi}_k^T\mathbf{\Sigma}_{\mathbf{X}_k|\mathbf{B}_k}^{-1}\mathbf{\Xi}_k\mathbf{B}_k \\
&- \frac{\lambda}{2}\mathbf{B}_k^T\mathbf{\Delta}^T\mathbf{\Sigma}_{\mathbf{B}_{k+1}}^{-1}\mathbf{\Delta}\mathbf{B}_k + \beta\mathbf{B}_k^T\mathbf{\Xi}_k^T\mathbf{\Sigma}_{\mathbf{X}_k|\mathbf{B}_k}^{-1}\mathbb{E}\,\mathbf{X}_k|\mathbf{X}_{-1} \\
&+ \lambda\gamma\mathbf{B}_k^T\mathbf{\Delta}^T\mathbf{\Theta}_{k+1}^T\mathbf{\Sigma}_{\mathbf{X}_{k+1}|\mathbf{B}_{k+1}}^{-1}\mathbb{E}\,\mathbf{X}_{k+1}|\mathbf{X}_{-1}. \qquad (33)
\end{aligned}$$

Now, $\mathbf{B}_k|\mathbf{X}_{-1} \sim \mathcal{N}(\mathbf{\Phi}\mathbf{X}_{-1}, \mathbf{\Sigma}_{\boldsymbol{\xi}_k})$, therefore upon comparing terms, we obtain the following self-consistent equations.

$$\begin{aligned}
\mathbf{\Sigma}_{\boldsymbol{\xi}_k}^{-1} &= \mathbf{\Sigma}_{\mathbf{B}_k}^{-1} + \beta\,\mathbf{\Xi}_k^T\mathbf{\Sigma}_{\mathbf{X}_k|\mathbf{B}_k}^{-1}\mathbf{\Xi}_k + \lambda\,\mathbf{\Delta}^T\mathbf{\Sigma}_{\mathbf{B}_{k+1}}^{-1}\mathbf{\Delta}, \\
\mathbf{\Phi} &= \mathbf{\Sigma}_{\boldsymbol{\xi}_k}(\beta\,\mathbf{\Xi}_k^T\mathbf{\Sigma}_{\mathbf{X}_k|\mathbf{B}_k}^{-1}\mathbf{\Sigma}_{\mathbf{X}_k\mathbf{X}_{k-1}}\mathbf{\Sigma}_{\mathbf{X}_{k-1}}^{-1} \\
&\quad + \lambda\gamma\,\mathbf{\Delta}^T\mathbf{\Theta}_{k+1}^T\mathbf{\Sigma}_{\mathbf{X}_{k+1}|\mathbf{B}_{k+1}}^{-1}\mathbf{\Sigma}_{\mathbf{X}_{k+1}\mathbf{X}_{k-1}}\mathbf{\Sigma}_{\mathbf{X}_{k-1}}^{-1}). \qquad (34)
\end{aligned}$$

Setting up the iterations, similar to Blahut-Arimoto equations, we obtain the following iterative procedure with $t$ as iteration index.

$$\begin{aligned}
\mathbf{\Sigma}_{\boldsymbol{\xi}_k}^{(t+1)} &= \left(\mathbf{\Sigma}_{\mathbf{B}_k}^{-1\,(t)} + \beta\,\mathbf{\Xi}_k^{T\,(t)}\mathbf{\Sigma}_{\mathbf{X}_k|\mathbf{B}_k}^{-1\,(t)}\mathbf{\Xi}_k^{(t)} + \lambda\,\mathbf{\Delta}^{T\,(t)}\mathbf{\Sigma}_{\mathbf{B}_{k+1}}^{-1\,(t)}\mathbf{\Delta}^{(t)}\right)^{-1} \\
&\overset{(a)}{=} \left(\beta\,\mathbf{\Sigma}_{\mathbf{B}_k|\mathbf{X}_k}^{-1\,(t)} - (\beta-1)\,\mathbf{\Sigma}_{\mathbf{B}_k}^{-1\,(t)} + \lambda\,\mathbf{\Delta}^{T\,(t)}\mathbf{\Sigma}_{\mathbf{B}_{k+1}}^{-1\,(t)}\mathbf{\Delta}^{(t)}\right)^{-1}, \\
\mathbf{\Phi}^{(t+1)} &= \mathbf{\Sigma}_{\boldsymbol{\xi}_k}^{(t+1)}(\beta\,\mathbf{\Xi}_k^{T\,(t)}\mathbf{\Sigma}_{\mathbf{X}_k|\mathbf{B}_k}^{-1\,(t)}\mathbf{\Sigma}_{\mathbf{X}_k\mathbf{X}_{k-1}}\mathbf{\Sigma}_{\mathbf{X}_{k-1}}^{-1}
\end{aligned}$$

$$+\lambda\gamma\,\mathbf{\Delta}^{T\,(t)}\mathbf{\Theta}_{k+1}^{T\,(t)}\mathbf{\Sigma}_{\mathbf{X}_{k+1}|\mathbf{B}_{k+1}}^{-1\,(t)}\mathbf{\Sigma}_{\mathbf{X}_{k+1}\mathbf{X}_{k-1}}\mathbf{\Sigma}_{\mathbf{X}_{k-1}}^{-1})$$

$$\overset{(b)}{=}\quad\mathbf{\Sigma}_{\boldsymbol{\xi}_k}^{(t+1)}(\beta\,\mathbf{\Sigma}_{\mathbf{B}_k|\mathbf{X}_k}^{-1\,(t)}\mathbf{\Phi}^{(t)}(I-\mathbf{\Sigma}_{\mathbf{X}_{k-1}|\mathbf{X}_k}\mathbf{\Sigma}_{\mathbf{X}_{k-1}}^{-1})$$
$$+\lambda\gamma\,\mathbf{\Delta}^{T\,(t)}\mathbf{\Sigma}_{\mathbf{B}_{k+1}|\mathbf{X}_{k+1}}^{-1\,(t)}\mathbf{\Delta}^{(t)}\mathbf{\Phi}^{(t)}\mathbf{\Sigma}_{\mathbf{X}_{k+1}}^{-1}\mathbf{\Sigma}_{\mathbf{X}_{k+1}\mathbf{X}_{k-1}}\mathbf{\Sigma}_{\mathbf{X}_{k-1}}^{-1}), \qquad (35)$$

where in $(a)$ and $(b)$ we have used the expansion of $\mathbf{\Xi}_k, \mathbf{\Theta}_{k+1}$, matrix inversion lemma (Golub & Van Loan, 1996), and the identity $\mathbf{\Sigma}_{X|Y}\mathbf{\Sigma}_{XY}\mathbf{\Sigma}_Y^{-1}=\mathbf{\Sigma}_X^{-1}\mathbf{\Sigma}_{XY}\mathbf{\Sigma}_{Y|X}^{-1}$, where $X, Y$ and $Z$ are Normal distributed random variables. We can also write the transition probability of the bottleneck variables, $\mathbf{B}_{k+1}|\mathbf{B}_k$ using Theorem 1 and the following useful expansion of the required KL-divergence term.

$$D_{KL}(p(\mathbf{X}_{k+1}|\mathbf{B}_k)||p(\mathbf{X}_{k+1}|\mathbf{B}_{k+1})) \quad = \quad c+\frac{1}{2}(\mathbb{E}\,\mathbf{X}_{k+1}|\mathbf{B}_k-\mathbb{E}\,\mathbf{X}_{k+1}|\mathbf{B}_{k+1})^T\mathbf{\Sigma}_{\mathbf{X}_{k+1}|\mathbf{B}_{k+1}}^{-1}$$
$$(\mathbb{E}\,\mathbf{X}_{k+1}|\mathbf{B}_k-\mathbb{E}\,\mathbf{X}_{k+1}|\mathbf{B}_{k+1}), \qquad (36)$$

where,

$$\mathbb{E}\,\mathbf{X}_{k+1}|\mathbf{B}_k \quad = \quad \mathbf{\Sigma}_{\mathbf{X}_{k+1}\mathbf{X}_{k-1}}\mathbf{\Phi}^T\mathbf{\Sigma}_{\mathbf{B}_k}^{-1}\mathbf{B}_k=\mathbf{\Upsilon}_k\mathbf{B}_k,$$
$$\mathbf{\Sigma}_{\mathbf{X}_{k+1}|\mathbf{B}_k} \quad = \quad \mathbf{\Sigma}_{\mathbf{X}_{k+1}}-\mathbf{\Sigma}_{\mathbf{X}_{k+1}\mathbf{X}_{k-1}}\mathbf{\Phi}^T\mathbf{\Sigma}_{\mathbf{B}_k}^{-1}\mathbf{\Phi}\mathbf{\Sigma}_{\mathbf{X}_{k-1}\mathbf{X}_{k+1}}.$$

After substituting equation (36) in (6) of Theorem 1, we obtain

$$\log p(\mathbf{B}_{k+1}|\mathbf{B}_k)=c-\frac{1}{2}\mathbf{B}_{k+1}^T\mathbf{\Sigma}_{\mathbf{B}_{k+1}}^{-1}\mathbf{B}_{k+1}-\frac{\gamma}{2}\mathbf{B}_{k+1}^T\mathbf{\Theta}_{k+1}^T\mathbf{\Sigma}_{\mathbf{X}_{k+1}|\mathbf{B}_{k+1}}^{-1}\mathbf{\Theta}_{k+1}\mathbf{B}_{k+1}$$
$$+\gamma\mathbf{B}_{k+1}^T\mathbf{\Theta}_{k+1}^T\mathbf{\Sigma}_{\mathbf{X}_{k+1}|\mathbf{B}_{k+1}}^{-1}\mathbf{\Upsilon}_k\mathbf{B}_k. \qquad (37)$$

Again, since $\mathbf{B}_{k+1}|\mathbf{B}_k \sim \mathcal{N}(\mathbf{\Delta}\mathbf{B}_k, \mathbf{\Sigma}_{\boldsymbol{\xi}_{k+1}})$, after comparing the terms on both sides of the equation (37), we obtain

$$\mathbf{\Sigma}_{\boldsymbol{\xi}_{k+1}}^{-1} \quad = \quad \mathbf{\Sigma}_{\mathbf{B}_{k+1}}^{-1}+\gamma\mathbf{\Theta}_{k+1}^T\mathbf{\Sigma}_{\mathbf{X}_{k+1}|\mathbf{B}_{k+1}}^{-1}\mathbf{\Theta}_{k+1},$$
$$\mathbf{\Delta} \quad = \quad \gamma\mathbf{\Sigma}_{\boldsymbol{\xi}_{k+1}}\mathbf{\Theta}_{k+1}^T\mathbf{\Sigma}_{\mathbf{X}_{k+1}|\mathbf{B}_{k+1}}^{-1}\mathbf{\Upsilon}_k. \qquad (38)$$

After setting up the iterations like we did for equation (34), and using the (18)-(20) we first write that

$$\mathbf{\Sigma}_{\mathbf{B}_{k+1}}^{(t)} \quad = \quad \mathbf{\Delta}^{(t)}\mathbf{\Sigma}_{\mathbf{B}_k}^{(t+1)}\mathbf{\Delta}^{(t)}+\mathbf{\Sigma}_{\boldsymbol{\xi}_{k+1}}^{(t)},$$
$$\mathbf{\Sigma}_{\mathbf{B}_{k+1}|\mathbf{X}_{k+1}}^{(t)} \quad = \quad \mathbf{\Delta}^{(t)}\mathbf{\Sigma}_{\mathbf{B}_k|\mathbf{X}_{k+1}}^{(t+1)}\mathbf{\Delta}^{(t)}+\mathbf{\Sigma}_{\boldsymbol{\xi}_{k+1}}^{(t)},$$
$$\mathbf{\Sigma}_{\mathbf{B}_{k+1}\mathbf{X}_{k+1}}^{(t)} \quad = \quad \mathbf{\Delta}^{(t)}\mathbf{\Sigma}_{\mathbf{B}_k\mathbf{X}_{k+1}}^{(t+1)}. \qquad (39)$$

Using (38) and (39), we obtain the following.

$$\mathbf{\Sigma}_{\boldsymbol{\xi}_{k+1}}^{(t+1)} \quad = \quad \left(\gamma\mathbf{\Sigma}_{\mathbf{B}_{k+1}|\mathbf{X}_{k+1}}^{-1\,(t)}-(\gamma-1)\mathbf{\Sigma}_{\mathbf{B}_{k+1}}^{-1\,(t)}\right)^{-1},$$
$$\mathbf{\Delta}^{(t+1)} \quad = \quad \gamma\mathbf{\Sigma}_{\boldsymbol{\xi}_{k+1}}^{(t+1)}\mathbf{\Sigma}_{\mathbf{B}_{k+1}|\mathbf{X}_{k+1}}^{-1\,(t)}\mathbf{\Sigma}_{\mathbf{B}_{k+1}\mathbf{X}_{k+1}}^{(t)}\mathbf{\Sigma}_{\mathbf{X}_{k+1}}^{-1}\mathbf{\Sigma}_{\mathbf{X}_{k+1}\mathbf{X}_{k-1}}\mathbf{\Phi}^{T\,(t)}\mathbf{\Sigma}_{\mathbf{B}_k}^{-1\,(t+1)}$$
$$\overset{(a)}{=} \quad \left(I-\frac{\gamma-1}{\gamma}\mathbf{\Sigma}_{\mathbf{B}_{k+1}|\mathbf{X}_{k+1}}^{(t)}\mathbf{\Sigma}_{\mathbf{B}_{k+1}}^{-1\,(t)}\right)^{-1}\mathbf{\Delta}^{(t)}\left(I-\mathbf{\Sigma}_{\mathbf{B}_k|\mathbf{X}_{k+1}}^{(t+1)}\mathbf{\Sigma}_{\mathbf{B}_k}^{-1\,(t+1)}\right), \qquad (40)$$

where $(a)$ is written using the previously used identity $\mathbf{\Sigma}_{X|Y}\mathbf{\Sigma}_{XY}\mathbf{\Sigma}_Y^{-1}=\mathbf{\Sigma}_X^{-1}\mathbf{\Sigma}_{XY}\mathbf{\Sigma}_{Y|X}^{-1}$, and matrix inversion lemma is used at each step. ∎

