# OpenReview forum: "Learning Information Propagation in the Dynamical Systems via Information Bottleneck Hierarchy"
_ICLR.cc/2019/Conference_

### Official Review · AnonReviewer3 · 2018-10-16
**ICLR 2019 Conference Paper576 AnonReviewer3**

**Rating:** 5
**Confidence:** 4

**Review:**

This paper studied an extension of the Information Bottleneck Principle called Information Bottleneck Hierarchy (IBH).  The goal of IBH is to extract meaningful information from a Markov Chain. Then the authors studied case of the Gaussian linear dynamic and proposed an algorithm for computing the IBH. Then an experiment was conducted to show the usage of IBH to practical problems.

Generally I like the idea of extending Information Bottleneck to dynamic systems and I think the experiment is interesting. But I have some major questions to the paper and these questions are important about the principle you are proposing.

1. About Figure 1, there is a link between X_{k-1} and B_k, but there are no link between X_k and B_{k+1}. I understand what you said --- B_k needs to compress X_{k-1} and delivers information to B_{k+1}. My question is ---- Figure 1 can not be generated to a longer Markov Chain. It seems that the principle you proposed only works for 3 random variables X_{k-1}-X_k-X_{k+1}, which weaken the principle a lot. Please draw a longer Markov Chain like Figure 1 to illustrate your principle.

2.  About the \epsilon_{1,2,3} in formula (3). \epsilon_1 is claimed to bound the accuracy of the prediction of X_k by B_{k-1}, but where not B_{k-1} appear in the formula (actually B_{k-1} is not even in Figure 1)? \epsilon_3 is claimed to define the closeness of prediction of X_{k+1} by B_{k+1}, but why does I(X_{k-1},X_{k+1}) need to be small? In the "Markov chains are considered" before formula (3), there are some typos, for example, X_{k+1}-B_k-B_{k+1} seems not a Markov Chain. Also why you are bounding the difference of two mutual informations, but not take the absolute value (I think the difference you are considered are not guaranteed to be non-negative)? I think formula (3) is the key to understand the IBH principle, but it is not well illustrated for the readers to understand.

3. I understand that you can only derive an algorithm for Gaussian linear dynamic, since non-Gaussian case might be difficult and Gaussian linear dynamic might be good enough for modeling real random processes. But I wonder what is the physical or practical meaning for the matrices \Psi and \Delta? Why \Delta can be used to predict sentiment intensity in your experiment? It seems that \Delta carries the information from B_k to B_{k+1}, so it is only one-hop information and the sentiment intensity involves multi-hop information. How do you combine the different \Delta for different hops to predict sentiment intensity? These questions are not well illustrated in the paper.

So I think the paper can be accepted if the author can provide some more insightful illustrations, especially for Figure 1, formula (3) and the experiment. But overall I think the idea in this paper is interesting, if well illustrated.

---

> ### Author Response · Authors · 2018-11-15
> **Response to Reviewer-3**
>
> We are extremely grateful for the careful reading of the manuscript and would like to thank the reviewer for appreciating the novelty of the proposed results. We also would like to clarify the concerns raised by the reviewer regarding the technical contribution and hope that we would converge by utilizing this discussion period. We now address the comments point-wise as follows.
>
> 1. We thank the reviewer for the suggestion of drawing the complete Markov Chain, and to explain the part of the problem that we are solving. We have modified the Figure 1 in the revised manuscript. We now like to mention that one of the main contributions of the paper is to write the IBH as an optimization problem. An advantage of such formulation is that it can be very well generalized to any number of hops (as mentioned in the manuscript, ending of Sec-3.1). In this work, we have introduced all these concepts and solved the problem for three variable case i.e. X_{k-1}-X_{k}-X_{k+1}. The IBH solution can be derived for the general case, by appropriately differentiating the information terms. For example, a generalized differentiation can be written as
> \begin{equation}
> \frac{\delta I({\bf B}_{k};{\bf B}_{k+1})}{\delta p({\bf B}_{j+1}\vert {\bf B}_{j})} = p({\bf B}_{j})(\mathbb{E}_{{\bf B}_{j+2}\vert {\bf B}_{j+1}}\ldots\mathbb{E}_{{\bf B}_{k}\vert {\bf B}_{k-1}}[D_{KL}(p({\bf B}_{k+1}\vert {\bf B}_{k})\vert\vert p({\bf B}_{k+1}))]-1),\quad\forall 1\leq j < k.
> \end{equation}
>
> The mathematical generalization of the IBH is already derived, and now we are looking for some interesting datasets which can benefit from larger than three hop dynamical systems. The generalization with new real-world datasets will be presented in the future as the full version.
>
> 2. We agree with the reviewer that the optimization problem (3) is critical to understand the IBH principle. Towards that, we would like to mention the following.
> a) There is a typo in the manuscript, what we meant to say is "\epsilon_1 bounds the accuracy of the prediction of X_k by B_k", since B_k is is the bottleneck corresponding to the first hop (X_{k-1}-X_k), therefore it is designed such that it better predict the X_k by compressing the information from X_{k-1}.
>
> b) The Markov chains are correct in the sense that, we design B_{k+1} such that all the information that B_{k+1} received is only through B_{k}, to construct an alternate dynamical process of B_{k}'s. In other words, this means that B_{k} is the parent of B_{k+1} (from a Bayesian network perspective) and hence B_{k+1} is independent of all other RV in consideration given B_{k}.
>
> c) The primary purpose of the constraints in the optimization problem (3) is to lower bound the prediction accuracies (to guarantee accurate predictions) and upper bound the compression at each hop of the alternate dynamical system (IBH). We note that only signs of the associated terms matter in this respect (as far as writing the Lagrangian is concerned), however, the mutual information constants in the constraints are carefully chosen by reverse-engineering from the proof of the Lemma 1. It has been shown in equation (21)-(22) that these differences are actually KL-divergence terms and hence guaranteed to be positive. This formulation also guarantees the lower boundedness of the functional in (4). Therefore, there is no need for the absolute difference in our framework as we only need the inequalities in one appropriate direction.
>
> 3. The reviewer is right in saying that \Phi matrix captures information flow from X_{k-1} to B_{k}, and \Delta is from B_{k} to B_{k+1}. We now note that B_{k} is designed to tap the information flow from X_{k-1} to X_{k} (here, from text to audio), and B_{k+1} wish to receive information from B_{k} to explain X_{k+1} (visual). Hence, \Delta has correlations from all three modalities because it receives correlations from text-audio and uses them to explain the visual modality. This is why \Delta is used for the sentiment prediction. A further explanation is added in the revised manuscript in the experiments sections.
>
> 4. Also, we have made a small cosmetic change of moving the Theorem 1 proof to the appendix.

---

### Official Review · AnonReviewer2 · 2018-11-02
**Preliminary, with 1 promising experiment, but unclear and vague**

**Rating:** 4
**Confidence:** 2

**Review:**

The paper proposes a method to learn the conditional distribution of a random variable in order to minimize and maximize certain mutual information terms.  Interestingly, the proposed method can be applied to sentiment prediction and outperforms a 2018 method based on SVM.

Overall, the ideas seem intriguing, and the results seem promising, but I really cannot understand what the paper is saying, and I think the paper would be much stronger if it was written more clearly (to make individual sentences more clear, but also to make the broader picture more clear). Not only is the writing hard to understand (some sentences lack a verb!), but it is vague, and the notion of a "complex system" is never defined.  It seems that the technique can be applied to any (potentially non-stationary) Markov process?

Additionally, due to the lack of clarity in the writing and lack of mathematical rigor, Theorem 1 does not seem to be true as stated. I think this is an issue of stating the assumptions, and not due to a mistake in the derivation.  Right now, the actual conclusion of theorem 1 is not even clear to me.

Quality: poor/unclear
Clarity: very poor
Originality: unclear, perhaps high? Not clear how related it is to the methods of Tishby et al.
Significance: unclear, as clarity was poor, and there was minimal discussion of alternative methods.

Specific points:

- Eq (2), the first term is included because it is for the "information compression task", but I do not understand that. Where is the actual compression?  This is not traditional compression (turning a large vector into a smaller vector), but more like turning one PDF into a PDF with lower entropy?

- This paper seems to fall into the subfield of system identification (at which I am not an expert), so I'd expect to see some related literature in the field. The only compared method was the IF method of Tishby et al. from 18 years ago (and the current work seems to be a generalization of that).

- Equation (4): what exactly is the object being minimized? Is it a PDF/probability measure? Is it an *instance* of a random variable?  If it is a PDF, is it the PDF of B_k | X_{k-1} ?

- The statement of Theorem 1 is either too vague or wrong. To say "The solution... is given by" makes it sound like you are giving equations that define a unique solution. Perhaps you mean, "Any solution ... must necessarily satisfy..." ? And that is not clearly true without more work. You are basically saying that any minimizer must be a stationary point of the objective (since you are not assuming convexity). It seems everything is differentiable?  How do you know solutions even exist -- what if it is unbounded? In that case, these are not necessary conditions.

- Lemma 1: "The iterative procedure... is convergent."  The iterative procedure was never defined, so I don't even know what to make of this.

- Section 3.2: "As proved by prior work, the optimum solution obtained by a stochastic transformation that is jointly Gaussian with bottleneck's input."  I do not know what you are trying to say here. There's no predicate.

- Section 4 wasn't that interesting to me yet, since it was abstract and it seemed possible that you make a model to fit your framework well. But section 5 is much better, since you apply it to a real problem. However, what you are actually solving in section 5 is unclear. The entire setup is poorly described, so I am very confused.

---

> ### Author Response · Authors · 2018-11-15
> **Response to Reviewer-2**
>
> We want to thank the reviewer for the detailed reading of the manuscript. We agree with the reviewer that the presentation of the novel results can be improved with further clarifications/modification of the text. With the unique opportunity provided by open-review and the discussion period, we are positive that all the raised comments and concerns can be addressed to further strengthen the manuscript. We have revised the paper, and now we further elaborate the comments as follows.
>
> 1. The details of the used compression term were first provided in the original IB paper (Tishby et a. 2000), where the authors have explained that the bottleneck variable (B) is determined via a stochastic mapping p(B|X) such that it produces a soft-clustering of the input variable (X). It has been realized that we are not looking at minimization of the entropy of the bottleneck variable (which is the lower bound on the number of bits needed to specify B in a codebook). Instead, we look for the expected number of bits needed to specify the new variable (B) in a codebook without confusion, i.e., a mapping from X to B, such that X's can be clustered to B (hence compression), and the clusters are distinguishable. In other words, this is equivalent to the number of B's in the codebook such that they represent distinguishable clustering of X, which is equal to vol(X)/vol(X mapped to the same B) = 2^I(X; B) using standard asymptotic equipartition property arguments.
>
> It can be noted that such compression can lead to spurious results, as we can very well choose to throw away all the details and reduce the required bits to zero (by mapping all of the input to a single cluster). So, the compression has to be done with some constraints of prediction (correlation) accuracies which appear through the second term in equation (2). The combined tradeoff is now a variational problem which is what is known as Information bottleneck problem, or equation (2). Further details are provided in the original IB paper.
>
> 2. The objective in equation (4) is a functional (function of functions), and the arguments are the associated probability distributions in the optimization problem (3). We have further clarified by explicitly writing the variables in the equation (4) in the revised manuscript.
>
> 3. In Theorem 1 what we meant to say is the conditions for the optimal solution (now revised). However, since we noted that the functional (4) might not be convex in the product space of the probabilities, we can only come up with a locally optimal solution.
>
> We have made some cosmetic changes in this section, by moving the Theorem 1 proof to the appendix (to get more space for explanation). We also specified clearly the iterative solution (which previously was a section in the Appendix) as Corollary 1. We also emphasize that in the proof of the Lemma 1 we have already proved that the functional is lower bounded. We also show that each iteration monotonically reduces the functional, hence we are guaranteed to find a solution since convergence is guaranteed.
>
> 4. The experimental section is clarified further in the revision. The goal of the multimodal sentiment intensity dataset is to predict the sentiment of the speaker using the three available modalities (text, audio and visual). We have mapped the modalities into a dynamical system and then used IBH to compress the high-dimensional data and identify the complex correlations across the modalities using the \Delta matrix.
>
> 5. The manuscript is revised to take care of the presentation of the paper, especially Introduction, Sec-3.1, Sec-3.2.

---

> > ### Comment · AnonReviewer2 · 2018-11-26
> > **Still confusing**
> >
> > I read the authors' response. It clarifies a few things, and they have revised the text, which I have skimmed, and it is better, but I think it still could use more clarification.  As most reviewers agree, the writing could be more clear, and you could discuss things more slowly, e.g., the IB background, then multimodal sentiment intensity (even in the revised version, there is still not nearly enough setup for this -- it is completely new to me, and to many other readers as well I would guess).
> >
> > The math is still sloppy in places, even if it's not necessarily wrong. For example, lemma 1 seems straightforward, and I have no issues with the proof technique (lower bound + monotonicity). To recall, it says: "Lemma 1. The iterative procedure in Corollary 1 to minimize the functional F in (4) is convergent."
> >
> > But then the text says "The convergence of the iterations in Corollary 1, and hence existence of a minimizing solution to (4), is established through the following result. ... [then states Lemma 1]"
> >
> > I take issue with the "hence existence of a minimizing solution". How did you infer this? Corollary 1 says nothing about *what* the sequence converges too. You could be minimizing the function x^2, and your sequence could be x_n = 1 + 1/n. This is monotonic and lower bounded, but it converges to x=1, not x=0.  I am guessing that the authors are aware of this, and it is just an issue of how the phrased things, but as written, this paper is still just too confusing.
> >
> > While I am not that excited about this version of the paper, I think the ideas and results are good, so I strongly recommend that the authors resubmit a longer, more clear version to a different venue.

---

> > > ### Author Response · Authors · 2018-11-27
> > > **Clarification**
> > >
> > > We would like to thank the reviewer for reading our revised version and for the feedback. We believe that the current version of the paper, in which we have incorporated a small change, clarifies the ambiguity regarding the convergence of the procedure.
> > >
> > > 1-     Due to space limitations, we have just restricted ourselves to a brief overview of concepts (IB and sentiment analysis) that have been introduced previously in the literature + citing all the most relevant papers that provide all the details.
> > >
> > > 2-    To recall, the idea of finding a solution is to use the self-consistent equations derived in Theorem 1 to develop the iterative procedure stated in Corollary 1. The convergence of the proposed procedure is claimed in Lemma 1. By construction, the iterative procedure is convergent to a stationary point since the self-consistent equations are derived by setting the derivatives of the functional (4) to zero. To avoid further confusion, we clarify this by explicitly writing it in the statement of Lemma 1. Indeed, if the procedure converges, then the point of convergence is necessarily a stationary point, that satisfies the self-consistent equations hence the derivatives of the functional (4) with respect to p(B_{k}|X_{k-1}) and p(B_{k+1}|B_{k}) are zero. We are confident that the proof of the convergence is correct, and we observe the same behavior in numerical as well as real-world results. We would also like to emphasize that the key idea in the developed proof is somewhat (but not directly) similar to the Expectation-Maximization (EM) technique, as mentioned by the cited works of Blahut-Arimoto and Tishby. In the original EM, the E- and M-steps can both be alternatively interpreted as the maximization of the objective (log likelihood), as observed in many existing works, for example, equation (7a), (7b) in [1]. Hence, one EM iteration is marginal maximization over two variables; (i) conditional probability distribution, and (ii) parameters of the model. In our case, one update of the algorithm in Corollary 1, involves maximization over four variables, p_1, p_2, \phi_1, \phi_2, as detailed in Appendix C, which is performed marginally as well.
> > >
> > > As noted in the well-developed literature of EM algorithm, the convergence is almost always to a stationary point with such marginal maximization (or minimization in our case) technique due to the self-consistent nature of the equations ([2] Section 3.4.1 Page-79). We also like to emphasize that in such techniques of marginal maximization over multiple variables (here probability distributions), it has been realized that in most of the cases, the stationary convergence point is indeed a local maximum (for EM), and thereby local minimum in our case. A small random perturbation at an arbitrary stationary point can lead the marginal maximization algorithm to diverge (Page-80, [2]). We agree that directly claiming local minimum in our case is optimistic and hence we rephrase the sentence to be more careful. We want to stress that this does not change any novel formulation, results, and proofs of the work.
> > >
> > > 3-    Finally, we would like to emphasize that the conditions presented in Theorem 1 are derived by equating the derivative to zero, as presented in the proof. Our iterative procedure is derived from Theorem 1, and it converges to a point where the derivatives are zero (or stationary point), since it performs minimization based on the gradient approach. Indeed, we are not just using the monotonicity and lower bound to claim the convergence to a stationary point but also, by definition, we use the structure of the problem where the self-consistent equations are derived from minimizing the functional in (4). Therefore, the example provided by the reviewer will not occur in our analysis as ‘1’ is not a stationary point of f(x) = x^2.
> > >
> > > [1] Sam Roweis and Zoubin Ghahramani, “An EM algorithm for identification of nonlinear dynamical systems,” 2000.
> > >
> > > [2] G. McLachlan and T. Krishnan, “The EM Algorithm and Extensions.” John Wiley & Sons, New York, 1996.

---

### Official Review · AnonReviewer1 · 2018-11-05

**Rating:** 5
**Confidence:** 3

**Review:**

This paper studies the problem of compactly represent the model of a complex dynamic system while preserving information. The method is based on the information bottleneck method. Basically, for a dynamic system whose states changing from X_{k-1}, X_k to X_{k+1}, the "information bottleneck hierarchy" method learns a variable B_k and B_{k+1} such that B_k predicts B_{k+1} well, B_k predicts X_k well, and B_{k+1} predicts X_{k+1} well, while minimizing the information of X_{k-1} contained in B_k.

In my opinion, this is a very interesting framework for representing and learning a dynamic system. The paper then considers simple examples on a linear model with Gaussian noise and show that the IBH method performs better than the one-BN method. The simulation and the experiments on real data all show very good performance (even with the simple linear Gaussian estimator).

The reason that I give such a rating is that of the confusing writing.
* In the abstract, it is unclear what the goal is. For example, the second and third sentence do not explain the first sentence: "the task is a crucial task".
* Introduction is also very confusing. It seems there is not a good logic connecting each sentence.
* The paper does not give a good survey of other methods performing similar tasks, e.g., the ones the paper are comparing to in the experiment section. Therefore, it is hard to compare or to understand why the previous methods are worse.
* Figure 2: one-BN is not well defined. How do you design the IB locally?

---

> ### Author Response · Authors · 2018-11-15
> **Response to Reviewer-1**
>
> We appreciate the reviewers' views on the novelty and usefulness of the proposed results. We are also thankful for the careful reading of the manuscript, and comments which have led us to improve the presentation of the results further. We provide clarifications as follows, and would like to emphasize that the raised concerns can be addressed by taking advantage of the discussion period.
>
> 1. The abstract and introduction were modified to present the original ideas and the problem statement better.
> 2. The experiments section is further elaborated to have prior works and further intuition of the developed approach. We wish to claim that the proposed approach (by design) can identify complex inter- and intra- dependencies across spatio-temporal states of the dynamical system. Hence, representing the modalities as a linear dynamical system, then allow us to come up with an alternate compact dynamical system which can be used to perform discriminative learning of the speaker sentiment using machine learning classifiers like SVM.
> 3. By one-BN, we mean designing an Information bottleneck between any consecutive pair of RVs in the given Markov Chain. The IBH designs series of bottlenecks jointly across the entire given dynamical system. In Figure 2, we try to emphasize that the joint design performs better than the local (or individually/independently designed) bottlenecks across the hops of the dynamical system. We elaborate this further in the simulation results section of the revised manuscript.
> 4. Also, we have made a small cosmetic change of moving the Theorem 1 proof to the appendix.

---

### Author Response · Authors · 2018-11-20
**Changes Summary**

We address the reviewers comments, in detail, individually in the responses, and a brief overview of the changes in the revised version are as follows:

1. The abstract and introduction are modified to reflect the problem setup and our proposed novel ideas better.

2. We further elaborate the Figure 1, thanks to the suggestions of the Reviewer-3

3. The constraints are further elaborated for the optimization problem in (3)

4. The proof of the Theorem 1 has been moved to the Appendix

5. The proposed iterative solution (initially present in the Appendix) is now termed as Corollary 1 for a better connection with the text of Section 3.1

6. The real-world experiment of sentiment prediction in Section 5 is further elaborated in terms of the problem setup, prior works. Further, we add more explanation of the applied techniques of the proposed work

---

### Meta-Review · Area_Chair1 · 2018-12-18

**Confidence:** 4
**Recommendation:** Reject

**Metareview:**

The reviewers reached a consensus that the paper is not ready for publication in ICLR. (see more details in the reviews below. )